# Odorant-Binding and Chemosensory Proteins in *Anthonomus eugenii* (Coleoptera: Curculionidae) and Their Tissue Expression

**DOI:** 10.3390/ijms24043406

**Published:** 2023-02-08

**Authors:** Pablo Lechuga-Paredes, Obdulia Lourdes Segura-León, Juan Cibrián-Tovar, Brenda Torres-Huerta, Julio César Velázquez-González, José Luis Cruz-Jaramillo

**Affiliations:** 1Colegio de Postgraduados, Campus Montecillo, Mexico-Texcoco Highway, Km. 36.5 Montecillo, Texcoco 56230, Mexico; 2Koppert de México, Circuito el Marqués Norte No. 82, Industrial Park el Marqués, Querétaro 76246, Mexico; 3Bioinformatics and Technologies Department, Solaria Biodata, Antonio Ortega 817, Benito Juárez, Mexico City 03100, Mexico

**Keywords:** perireceptor proteins in olfaction, gene expression pattern, pepper weevil, RNA-seq, RT-PCR

## Abstract

The pepper weevil *Anthonomus eugenii* is one of the most damaging pests to the pepper crop. To offer alternative management strategies to insecticides, several studies have identified the semiochemicals that are involved in the pepper weevil’s aggregation and mating behavior; however, there is no information on its perireceptor molecular mechanism, to date. In this study, bioinformatics tools were used to functionally annotate and characterize the *A. eugenii* head transcriptome and their probable coding proteins. We identified twenty-two transcripts belonging to families related to chemosensory processes, seventeen corresponding to odorant-binding proteins (OBP), and six to chemosensory proteins (CSP). All results matched with closely related Coleoptera: Curculionidae homologous proteins. Likewise, twelve OBP and three CSP transcripts were experimentally characterized by RT-PCR in different female and male tissues. The results by sex and tissue display the different expression patterns of the AeugOBPs and AeugCSPs; some are present in both sexes and all tissues, while others show expressions with higher specificity, which suggests diverse physiological functions in addition to chemo-detection. This study provides information to support the understanding of odor perception in the pepper weevil.

## 1. Introduction

*Anthonomus eugenii*, commonly known as pepper weevil, is one of the pests that limit pepper production in Mexico and North America [1,2,3]. The presence of the weevil potentially causes complete crop losses, since the insect completes its life cycle in the fruits [4]. It presents more than one generation per year; in the absence of pepper plants, it can remain in other wild Solanaceous until it finds a more suitable host; pepper-weevil management has become a laborious and costly activity [5].

Management strategies include broad-spectrum pesticide applications; however, the effectiveness is diminished by the biology of this insect, and there are reports on the development of resistance mechanisms, which makes its control even more difficult with the use of a single strategy [3]. Therefore, there is particular interest in developing and integrating different management tactics, one of the most promising is the use of an aggregation pheromone [2,6,7]. However, so far, it has only been used for monitoring and the research in this area is attempting to obtain new formulations that assist the pepper-weevil control [8,9,10,11]. 

Non-receptor proteins are part of the insect perireceptor system, which includes pheromone recognition. They play essential roles in insect behavior and physiology such as foraging, mating and oviposition [12]. Consequently, there has been more interest in describing and characterizing the functional components involved in the insects’ perception process and the semiochemical recognition of pests such as *A. eugenii*, to develop tools to be incorporated as management tactics [13]. Two families of proteins that are part of the peripheral nervous system of the Insecta class are odorant-binding proteins (OBP) and chemosensory proteins (CSP), whose primary function is recognition and odor-molecule volatile transport to specific receptors [14,15]. 

OBPs are small globular proteins of 100–160 amino-acid residues, with a fold pattern in six α-helices forming a cavity with an affinity to hydrophobic substrates [16]. Different subfamilies have been characterized based on their protein primary structure and conserved cysteine (Cys) patterns; the OBP Classic with six Cys conserved [14]; the Plus-C subfamily with more than six Cys and a conserved proline; the Minus-C subfamily with 4–5 Cys [17,18], and dimers [19]. While CSPs are a family of highly conserved small proteins with 100–150 amino-acid residues [20], they have four conserved Cys residues which bind at two disulfide bridges and form a channel for transporting molecules [12,21].

Through Next-generation RNA-Seq (RNA sequencing) and the development of bioinformatics tools, knowledge of the genes involved in diverse signaling and metabolic pathways in non-model insect species has expanded. These approaches provide a way to identify and annotate de-novo genes encoding homologous proteins to those described in biological databases, and infer similar functions [22,23]. The above has enabled the understanding of the genetic basis of chemosensory perception in insects and the selection of targets with potential for the development of new technologies for pest management [24]. In this study, by analyzing the transcriptome of male and female *A*. *eugenii* heads, we identified and characterized the OBP and CSP chemosensory families, which are related to the recognition and transport of odors. In addition, using RT-PCR, we validated the identified genes and obtained their expression profiles in five different tissues of adult males and females.

## 2. Results

### 2.1. Library Assembly

As a result of sequencing two Illumina MiSeq-paired libraries of 25 male and 25 female heads of *A. eugenii*, we obtained a total of 9.426447 million clean reads with seventy-five base pairs (bp) average length, Q20 percentage of 96.33%, and GC percentage of 34.14% (Appendix A). The reads from the two paired libraries were assembled in 7130 sequences, with a mean length of 356 bp and N50 length of 647 bp. After cleaning redundancies, 5855 unigenes were obtained, with a mean length of 327 bp and N50 of 538 bp (Appendix A).

### 2.2. GO Terms, Homology Annotation and Chemosensory Sequences Extraction 

For transcript annotations, the Gene Ontology (GO) database was used; 2194 transcripts were annotated from 5855 unigenes in at least one of the three GO terms (Appendix A). Molecular functions were the most representative with 1774 genes, corresponding to binding functional group (44.5%) and catalytic activity (36.5%). A total of 1352 genes were annotated as Biological Process, with metabolic processes (28.4%) being the most representative subcategory, followed by cellular processes (25.8%), localization (9.7%), and biological regulation (7.8%). Finally, 883 were assigned Cellular Components, where the most represented subcategories were cell parts (17.1%) and cells (17.1%) (Figure 1). 

On the other hand, we used the Insecta-UniProtKB database for the unigenes homology annotation. Coleoptera was the order with the best hits, *Dendroctonus ponderosae* (31.8%) and *Tribolium castaneum* (23%) had the highest representation followed by other species, including *Anthonomus grandis*, the boll weevil. The identity percentage was from 40–100% in 89% of all annotated genes, and most E-values were >1 × 10^−50^–1 × 10^−100^ (Figure 2).

The homology and GO annotations allowed the identification and filtering of non-receptor genes, and the conserved domain database (CDD) confirmed the functionality of each transcript. Finally, we used the DescribePROT database based on model organisms to classify the transcripts (Appendix A). After analysis, seventeen encoders AeugOBPs and six AeugCSPs were obtained. In addition, one putative partial gene was identified by mapping to *A. grandis* CSPs as a reference index and corroborated by the OS-D domain presence.

### 2.3. Odorant-Binding Proteins

We identified seventeen transcripts encoding OBPs with lengths from 70 to 153 residues; eleven had complete ORFs and six incompletes, two of each in the 3’, intermedium, and 5’regions. The BLAST match with OBPs with 28.46–87.97% percentage identity, mainly with a member of the same genera A. grandis, followed by others Coleoptera Curculionidae: D. ponderosae, Ips typographus, and Rhynchophorus ferrugineus with identities more significant than 50%, except for AeugOBP83b, which was 28.4%, (Appendix A). Meanwhile, in DescribeProt, fifteen AeugOBPs had matches with OBP from T. castaneum and two with *Drosophila melanogaster*; these results were used to name A. eugenii transcripts. The conserved domain CDD/SPARCLE and InterPro databases assigned all AeugOBPs in the insect pheromone/odorant-binding proteins superfamily (CDD cl11600/PF01395, InterPro/SSF47565) and fourteen within the pheromone-binding protein–general odorant-binding proteins (PBP/GOBP smart00708) (Table 1). 

Two OBP subfamilies were annotated in *A. eugenii* based on the conserved Cysteines (Cys) pattern. Seven correspond to the classic OBP, with six conserved Cys and pattern C1-X_23_-44-C2-X_3_-C3-X_36–43_-C4-X_8–12_-C5-X_8_-C6 (Appendix A), and ten assigned to the Minus-C, with four conserved Cys and the absence of C2 and C5 (Appendix A). The motif analysis from A. eugenii transcripts and the best hits in BLASTx resulted in ten motifs with different architectural distributions in each subfamily (Appendix A). 

The motif analysis from A. eugenii transcripts and the homologs from BLASTx resulted in ten motifs with different architectural distributions in each subfamily (Appendix A). Among the classic OBP, the most conserved motifs were 2, 3, and 5, followed by 1, 6, and 4. Motifs 1 and 2 have Cys-X_3_-Cys and Cys-X_8_-Cys patterns corresponding to the classic OBPs. Three architectures were the most frequent, one with ten sequences including OBP24 and OBP6, another with seven sequences containing OBP14, and the other without any A. eugenii. Two AeugOBPs motif architectures were unique: OBP73a and OBP83b (Appendix A). 

In contrast, the minus-C subfamily displayed a different motif architecture, with high motif conservation where motifs 1, 2, 5, and 6 contain the conserved Cys residues characteristics of the minus-C subfamily. The presence of motifs 8 and 3 splits the architectures into two main groups; motif 4 takes a different position in both groups (Appendix A).

Finally, a phylogenetic tree was constructed to understand the relationship between the putative AeugOBPs and their orthologues of thirteen beetle species (Appendix A). The best evolutionary model was WAG + I + G4, according to the Bayesian information criterion (BIC). The best tree showed two clades, one with the classic OBP and the other with minus-C OBP, with values > 0.9 of posterior probability (Figure 3). Most of the minus-C AeugOBP clustered with A. grandis proteins, in contrast with just three of the seven classic AeugOBP being clustered with boll weevil’s proteins. Finally, the motif architecture results added to the phylogenetic tree display a positive correlation between both analyses.

### 2.4. Chemosensory Proteins

We identified six CSPs with lengths from 80–146 residues and identities greater than 60% with *A. grandis, D. ponderosae*, and *Acromyrmex echinatior*. The ORF search indicated that two AeugCSPs were complete and four partial (Appendix A). The AeugCSP nomenclature was also obtained by homology with the model organism *T. castaneum* using the DescribePROT server with identity values over 50%. The functional annotation generated through CDD/SPARCLE database classified the AeugCSP within the insect pheromone-binding family A10/OS-D corresponding to cl04042 in CDD, and PF03392 in Pfam databases (Table 2).

The multiple alignments with homologous sequences displayed the four characteristic Cys patterns described for this protein family (C1-X_6_-C2-X_18_-C3-X_2_-C4) (Appendix A). Motif enrichment analysis resulted in eight conserved motifs. Motifs 1, 3, and 4 represent the four conserved Cys, and other highly conserved residues in all CSP of the coleopteran species analyzed, which contrasts with the high variability observed in the OBP (Appendix A). Two architectures were the most frequent, differing in the presence of motif seven.

The CSP phylogeny reconstruction was performed with Bayesian approximation with the JTT + G4 initial substitution model, selected by the BIC criterion. In the phylogenetic tree, the AeugCSP clustered with their respective orthologous AgraCSP with >90 posterior probability values (Appendix A). Integration of the motif architectures in the phylogenetic tree showed complementarity between the two results, with proteins presenting highly conserved AeugCSP motifs and their homologs in internal clades, whereas CSP lacking motif seven were established as a more ancestral group (Figure 4).

### 2.5. OBP and CSP Expression Profile 

The experimental protein validation was performed by designing specific oligos for the AeugOBP and AeugCSP and their amplification in five tissues from male and female and a pool of both sex heads as a positive control (PC). Twelve AeugOBPs and three AeugCSPs were amplified by RT-PCR (Figure 5); all amplicons had the expected molecular weight, length, and nucleotide sequence corresponding to the transcript identified in silico in the head’s transcriptome. The expression profile on PC showed more expression in some genes, whereas in the complete heads separated by sex (CH), AeugOBPC04a, OBPC04c, OBPC10a, OBP9, OBP10, OBP14, OBP24, CSP11, and CSP14 were expressed in both sexes. The AeugOBP04b, OBPC09, OBP73a, and OBP83b were amplified mainly in females. These results suggest differences in chemoreception processes between sexes.

On the other hand, expression in heads without antennae (IH) was similar in both sexes, where AeugOBP04c, OBP9, OBP24, CSP11 and CSP14 decreased in expression and CSP10 were only expressed in males. The contrast between CH and IH suggests the specific expression of genes in the antennae, whose reduced expression or non-amplification in IH may be attributed to their absence. Those results support the antenna as an active site of those genes. Likewise, the thorax (T) and abdomen (A) showed fewer amplified and lower expressed genes in both sexes. Finally, gene expression in the legs (L) showed that all classical OBP and CSP11 are highly expressed in both sexes, except for OBP9, which is only present in females. On the other hand, OBP-minus reduced its expression, OBPC09 was not expressed, and CSP11 and CSP14 have similar results (Figure 5).

In addition to the differences obtained between tissues and sex, gene expression profiling also revealed possible multifunction, e.g., two Classic OBP (OBP10 and OBP73a) and one minus-C (OBPC10a) were expressed in all tissues evaluated, suggesting functions in addition to olfaction. Likewise, genes such as AeugOBPC04b and CSP11, which were amplified in CH and A, could correlate between olfaction and oviposition. Finally, the gene expressed in PC and L suggests that legs are an important chemosensory organ in peeper weevils, such as AeugCSP10, which was exclusively in the legs of both sexes.

## 3. Discussion

This work represents the first step towards understanding the proteins involved in the chemical communication in *A. eugenii*, by head transcriptome analysis and gene validation in five different tissues. The head transcriptome assembled resulted in 37% annotated unigenes against GO terms. The molecular function was the best represented (1774 unigenes); from these, 47% correspond to binding activity and 39.1% to catalytic function. Transcripts were predominantly annotated with sequences from *D. ponderosae*, T. castaneum and other related Coleopteran species due to their close evolutionary relationship with *A. eugenii* and the access to their genomic data [24]. Similar results have been obtained in other coleopterans studies, where antennae or complete head tissues were sequenced [25,26]. 

The OBP and CSP are central proteins involved in perireceptor events in odor recognition located in the sensilla, the primary olfactory unit of insect antennae [27]. Identifying both protein families in transcriptomic data is possible using different genomic databases and bioinformatics tools to search patterns present in proteins experimentally characterized in the uncharacterized proteins, in order to predict their possible function and origin [23,28]; the identities with homologous sequences of model organisms were around 30% [29]. OBP and CSP are characterized by their primary protein structure and conserved Cys patterns, with characteristic functional motif structures in each subfamily, and highly conserved patterns among insect orders [27,29].

Transcriptome analyses have been used to identify and functionally characterize non-receptor proteins in other Coleopteran insects [26,30,31]. The conserved protein domains represent stable evolutionary units directly related to their tertiary structure, and the function in a protein family [32]. Hence, the criterion to identify AeugOBP and AeugCSP transcripts was through homology search, functional annotation domains, and phylogeny analysis, criteria that have already been used in another research [26,33].

We identified 17 AeugOBPs, all belonging to the PBP-GOBP superfamily (cl11600)/PF01395), of which 14 are included in the PhBP family (smart00708). The gene annotation was based on the best hit with its homologous gene from model organisms *T. castaneum *and *D. melanogaster*. Since there is no consensus on chemosensory genes annotation in the databases, and the best hit in BLAST can correspond to a different gene namea, we suggested standardizing genetic information using model organisms [34]. This annotation results in an advantage for functional annotation and origin; for instance, AeugOBP73a had 47.37% identity with DmelOBP73a isoform c, which is significant to the identity previously mentioned to establish homology [27]. Therefore, it is possible to infer its origin and evolution since this protein in *D. melanogaster* is one of the clear orthologous protein whiting 12 *Drosophila* genomes and it is present in almost all insects except in Hymenoptera [35,36]. 

From the two AeugOBP families identified based on their Cys, the minus-C is more diverse. In both families, the motifs with Cys residues are the most highly conserved in each group, not only in the sequences but also in its motif architecture position, among other amino acids, which are also conserved. The minus-C group was divided into two groups by a single change in the position of motif 8 to motif 3. In each group, motif 4 was highly conserved; however, in two of the eleven architectures, it took up different positions (Appendix A). These architectural changes in highly conserved motifs are also present in Classic and Minus OBP in *Dendroctonus adjunctus* [26].

The AeugOBPs phylogeny clustered closely related sequences from fourteen Coleoptera species. In general, AeugOBPs were grouped together with their sister taxa *A. grandis*, Curculionidae species of the genus *Dendroctonus*, and the model specie *T. castaneum*. Classic and minus-C OBP were distributed in distinct external clades, associated with their motif architecture and primary sequence lengths. A similar pattern has been reported in other Coleoptera Curculionidae species [26,37], which contrast with arthropod phylogenies, where no clear difference was reported, and polyphyletic origin was suggested to minus-C OBP [35]. All AeugOBP were clustered in different clades, except for OBPC01 and OBPC09, which were together; this contrasts with the AgraOBP minus-C subfamily, where more than one sequence was set in the same branch. 

In the phylogenetic analysis, the classical AeugOBP73a were part of the root clade from which there was a division into one Classical and two minus-C groups; this topology was reported in Coleoptera, Neuroptera [26,38] and *D. melanogaster* [39]. This clade also includes one of the most frequent motif architectures in Classical-OBP (motif 7), present in *T. castaneum* and other Curculionidae, such as *Ips typographus*, *D. ponderosae,* and *D. adjunctus* (Figure 3), which is congruent with OBP *T. castaneum* phylogeny. In addition, OBP73a is one of the two OBPs members with a clear orthologous relationship with high conservation in a large species of insects [35], suggesting a critical function for this protein [40]. Furthermore, this protein was expressed in the five evaluated tissues, indicating different roles in *A. eugenii* physiology.

The minus-C group was divided into two main clades; one includes the four OBPC04s; in the homology analysis all matched with *T. castaneum* reference genome (UniProtKB/D6WS38) with similar e-value, sharing motif 8 in its motif architecture. In addition, phylogenetic analysis clustered all OBPC04 proteins together, so their gene origin could be due to tandem duplication, comparable with *D. melanogaster* [39] and *T. castaneum* [30]. In addition, its profile expression shows a different expression in CH on both sexes; OBPC04a, OBPC04b, and OBPC04c were also expressed in the abdomen; those results suggest a functional diversification, as reported in *D. melanogaster *[39]*, T. castaneum* [29,30], and other OBPs genes in Arthropoda [35].

Meanwhile, the AeugOBPC10b clustered with four *A. grandis *proteins: OBP21, OPB7, OBP15 and OBP18, with similar motif architectures, where the highly conserved motif 4 occupied a different position compared to the other minus-C, which suggests a functional innovation in this genus. Others are OBPC04 and OBP10, which in* T. castaneum* have been in defense mechanisms against entomopathogenic fungi, mainly in wild populations [41]. In the pepper weevil, the OBPC04c, OBP10, and OBPC10a were expressed in all evaluated tissues, so its physiological functions could be like those described in *T. castaneum*. Previous studies in other Coleoptera help us to infer possible roles for AeugOBP based on their distribution pattern; however, future studies will help confirm a role in *A. eugenii* physiology.

The six CSPs have four conserved Cys residues, a different motif distribution pattern to OBP-minus, and higher conserved residues than OBP [35]. AeugCSP were classified in the insect pheromone-binding family A10/OS-D, and the best hit was also with *A. grandis *sequences, followed by *D. ponderosae*, *A. echinatior* (Hymenoptera), and with model organism *T. castaneum* with identity values >50%. The motif analysis resulted in seven architectures; two were the most frequent: one with 23 sequences, including AeugCSP1a and AeugCSP10, while the second had seven CSP, including AeugCSP14; the main difference between both structures was the absence/presence of motif 7 (Appendix A). The higher conservation of CSP compared to OBP has been attributed to a lower binding specificity for environmental volatiles, rather than the type of semiochemical involved in communication [14].

The phylogenetic tree showed each AeugCSP in a specific clade along with its orthologous sequences from the sister species *A. grandis*, most of them with a 100% posterior probability. Furthermore, the motif patterns matched the phylogeny where motif 7 absence was set in root branches, which suggests that motif 7 appeared later in the CSP evolution; similar findings were reported in *D. adjuntus* CSP [26]. However, these results contrast with the CSP phylogenies in other Curculionidae, such as *I. typographus* and *D. ponderosae*, where this family has divided into two different clades [42]. 

Thus far, most of the OBP and CSP information in non-model insects come from bioinformatics analysis; however, functional characterization by experimental methods is gaining relevance [37,43]. We amplified 12 OBP genes, six OBP Classical, six minus-C and three CSP in a head pool and five tissues of both sexes by RT-PCR. The male and female gene expression in complete heads showed a different expression pattern: 14 genes were expressed in females, while 12 genes were expressed in male heads. Similar sex-related patterns were observed in *A. grandis* [31]; results may be associated with the neuron morphology, where ten neurons in females responded to male-produced pheromones and plant-volatile odors, compared to only four neurons in males [44]. 

In general, the antennae elimination (IH) showed no differences in expression profile between males and females, except for AeugOBPC01, which had no expression, and AeugOBPC04b, which showed reduced expression. Conversely, AeugCSP10 was expressed only in males, while AeugCSP11 and –AeugCSP14 were not expressed. The difference between gene expression in the CH and IH suggests that genes present in the antenna are regulated in both sexes, except for AeugCSP10, which had an interesting expression pattern in the leg. In addition, the IH expression suggests that these proteins are not exclusive to antennae and can be expressed in gustatory tissues such as the maxillary palps and proboscis, important appendages in insect heads [36,39].

On the other hand, the reduction in gene number and expression in the thorax and abdomen contrasts with other regions. This result supports the idea that those genes can be involved in another function in no-sense organs, such as the hindgut, Malpighian tubule, fat body, seminal receptacle, and accessory glands [14,36]. Finally, eleven genes were present in legs (L); six correspond to classical AeugOBP, with a high expression level (OBP9, OBP10, OBP14, OBP24, OBP73, OBP83b) and four to minus-C (OBPC01, OBPC04a, OBPC04c, OBPC10a) with a lower expression than OBP-Classic, and AeugCSP10. In addition, we observed different expression patterns according to sex: eleven were present only in females and nine in males. These results support that the legs are an essential site of action of these genes and suggest contact chemoreception for non-volatile compounds. A similar result was reported in *A. grandis* by Paula et al. (2018), who suggested these results could indicate the importance of this gene in contact chemoreception [31].

The high expression level in the head with antennae and legs also supports that OBP and CSP are involved in the peripheral process as semiochemical carriers [14,15]. The higher expression of genes from both families in the CH and L of females than in males in pepper weevils is similar to that reported in the cotton weevil *A. grandis* by Paula et al. (2018), who also mentioned that female expression is modulated by semiochemicals exposition. It is also known that *A. eugenii* males modulate the production of aggregation pheromones depending on the density of other males [7], so these proteins could have a role in the weevil physiology for the modulation and production of aggregation pheromones. 

Genes with differential expression by tissue and sex could be involved in specific functions; for example, AeugOBPC04a, AeugOBP9, and AeugOBP24 were mainly expressed in CH, and their expression decreased in IH. Therefore, these genes could be essential in *A. eugenii* odor chemoreception through antennae and may regulate different behaviors such as foraging, mating and oviposition. This function has already been studied in other insect species, and by silencing the expression of specific genes, some physiological functions have been interrupted, for example, the search for oviposition sites [45]. Likewise, OBP83b of *D. melanogaster* has been reported in the odor stimuli deactivation process [46]. In the PCR analyses, the expression of the homologous AeugOBP83b gene was higher in females’ CH, and lower in IH, so this gene could be associated with antennae tissue and have a similar function to that of *D. melanogaster.*

The olfactory role of CSP has been confirmed since the first works that characterized this family of proteins [18,35], which have increased in the last decade [45,47,48,49,50,51]. For example, CSP1 in *Bombyx mori* is highly related to the chromophore retinal in the eyes [52]. Likewise, CSP2 in *Bemisia tabaci* showed an affinity for fatty acids and are involved in detoxification processes [53]. The last process was verified by RNAi in CSP10 in *T. castaneum *[54], whose homolog in *A. eugenii* (AeugCSP10) was only expressed in IH males and T/L of both sexes. This might indicate a relation between head and legs, and it would be interesting to evaluate the role of AeugCSP10 in the detoxification process in *A. eugenii* populations that have developed resistance to insecticides [3].

The AeugCSP11, another gene with differential expression in CH, IH and A, shows a reduction in the expression in the last two tissues. Therefore, it could also be a central protein in the chemoreception of odors. The expression of this gene in the abdomen can be associated with the oviposition process, as is suggested for OBP and CSP proteins in the terminal part of the abdomen [45]. However, more studies are required to confirm the gene function.

The present results provide the first step in the knowledge of non-receptor chemosensory proteins involved in the chemical communication of *A. eugenii*, whose behaviors such as host localization and mating are influenced by pheromone aggregation compounds and plant volatiles [7,8,9,10]. One of the main objectives in the study of these protein families is to understand their role in insect physiology and to find target genes for the development of strategies to be incorporated in integrated pest management, such as the discovery of novel semiochemicals (super-ligands) using homology modelling, the construction of biosensors and the manipulation of behaviors through gene silencing or gene editing [27,45,55,56,57].

Initially, OBP and CSP were characterized as peripheral olfactory proteins [14,15]. However, after 40 years of study, both protein families have been reported not only in primary sensory organs, with additional functions to chemoreception, related to nutrient transport, toxic-molecule capture, pheromone detection and release in specialized glands [14,36,58]. In addition to characterizing *in sillico* the OBP and CSP of *A. eugenii*, our results increase the information on their expression profiles in different tissues and sexes, which enables new interrogations to elucidate specific functions, with emphasis on the detection of semiochemicals that will help in the management of the pepper weevil.

## 4. Materials and Methods

### 4.1. Insect Collection, cDNA Library Construction, Sequencing, and De Novo Assembly

Adults of *A. eugenii* were collected from infested jalapeño pepper plants (*Capsicum annum*) in October 2018 in the Ejido Vallejos in Villa de Guadalupe, San Luis Potosí, Mexico. The insects were sorted by sex into groups of twenty-five individuals, and preserved in RNAlater Storage Solution (Thermo Fisher Scientific, Whaltham MA, USA) to keep the RNA stable, according to the manufacturer’s instructions. Two total RNA extractions from 25 male and 25 female heads were performed with the SV Total RNA Isolation System kit (Promega, Madison, WI, USA), and RNA quality and concentration were confirmed in a Qubit Flex Fluorometer (Invitrogen, Carlsbad, CA, USA). Afterward, we constructed two paired libraries for each RNA extraction with the TruSeq™ RNA Sample Preparation Kit v2 (Illumina, San Diego, CA, USA), according to the manufacturer’s instructions. Finally, the sequencing of two paired libraries was performed on a MiSeq™ System Illumina (San Diego, CA, USA) at the Massive Sequencing Laboratory at the Centro Nacional de Referencia Fitosanitaria of SENASICA in Tecámac, State of Mexico (Appendix A). Library quality were verified with FastQC v.0.10 (https://www.bioinformatics.babraham.ac.uk/projects/fastqc/ accessed in 15 July 2020), and adapters and low-quality reads were removed using the software Trimommatic [59]. The de-novo assembly was performed with the two paired libraries with default parameters in Trinity v 2.0.6 [60] at the Molecular Evolution Laboratory, Colegio de Postgraduados, Texcoco, State of Mexico. The assembling quality was verified with QUAST v. 5.0.2 [61], and redundancies were removed.

### 4.2. Gene Ontology (GO Terms) Annotation

The annotation of GO terms was performed by the mapping routine of HMMER2GO v.0.17.9 [62] against a local Pfam database and the results were visualized in WEGO 2.0 [63]. Homology analysis was performed with BLASTx against a dataset constructed with the Insecta database from UniProtKB with E-value 1 × 10^−6^. Finally, all unigenes from the *A. eugenii* transcriptome encoding OBP and CSP proteins were filtered. Additionally, a mapping routine was developed with Geneious Prime 2022 (Dotmatics, Boston, MA, USA) using standard parameters, with the goal of finding new CSP by performing an index with a sister species (*Anthonomus grandis*) as a reference to find any additional genes not identified with BLASTx.

### 4.3. Classification of OBPs, CSPs, and Functional Annotation

The identification of the open reading frame (ORFs) was performed with TransDecoder (https://github.com/TransDecoder/TransDecoder accessed on 15 January 2021), and the results were corroborated in ORFinder of NCBI [64]. Then, protein sequences were subjected to further homology analysis using BLASTp against the non-redundant protein sequences database of the NCBI, and the five results with the highest percentage of similarity and e-values greater than 1 × 10^−6^ were selected. Only sequences corresponding to Coleoptera were filtered and redundancies were removed. Then, multiple alignments by MUSCLE [65] were performed for each protein family to identify the conserved Cys patterns, and edited in GenDoc. Additionally, a paired alignment on the model organism database DescribePROT server [66] was carried out for *A. eugenii* OBPs and CSPs nomenclature assignation. Finally, for functional characterization, we searched conserved domains architecture in InterProScan [67] within the PFAM, SUPERFAMILY, CATH-Gen3D, and PANTHER databases, and a corroboration was developed in CDD/SCPARCLE of NCBI.

### 4.4. Motif Discovery and Phylogenetic Analysis

The motifs discovery was performed in MEME 5.4.1 [68] with the previously selected proteins in three sets—two for OBP divided by classical and minus-C subfamilies, and one for CSP—with the following parameters: minimum length = 6, maximum = 12, with a maximum number of 10 motifs for OBPs, and 8 motifs for CSPs; in all cases, the motifs with *p* < 0.0001 were selected. Phylogenetic analyses were conducted with the previously aligned sequences of each family, and the best substitution model for each family was obtained with ModelTest-NG [69]. The phylogenetic trees for OBP and CSP were built by Bayesian inference in BEAST v1.10.4 [70] with 4,000,000 MCMC generations. The phylogeny of each family was visualized and edited in iTOL v5 [71], and motif results were integrated.

### 4.5. Gene Amplification of OBP, CSP, and Expression Pattern per Tissue and Sex 

Jalapeño peppers infested with *A. eugenii* were collected in the Ejido Soledad de Graciano Sánchez, in San Luis Potosí, Mexico in October 2020, and transported to Colegio de Postgraduados Campus Montecillo for sample processing. The peppers were placed in plastic boxes until the adult emergence, which were placed in plastic cages and fed with fresh peppers for 5–8 days until they reached sexual maturity [1]. In this period, 100 females and 100 males were collected during their aggregation behavior (between 10:00–12:00 h) in 2mL Eppendorf tubes with RNAlater (Thermo Fisher Scientific) and were preserved following manufacturer’s instructions.

For the expression-profiles analysis, we dissected five different tissues from 50 males and 50 females: complete heads, heads without antennae, legs, thorax, and the final part of the abdomen. In addition, 100 additional male and female weevils were used to dissect the heads without antennae, and we included a positive control consisting of 30 complete heads from males and females (1:1). Total RNA extraction was made with the SV Total RNA Isolation System kit (Promega, Madison, WI, USA), and RNA quality and concentration were evaluated on a Thermo Scientific™ NanoDrop™ 2000. The cDNA was synthesized from 1µg of total RNA with the SuperScript™ II reverse transcriptase enzyme (Invitrogen™) with oligos (dT) 12–18 following the manufacturer’s instructions. Finally, cDNA concentration was checked with the NanoDrop™ 2000 (1060.6–5421 ng/μL), and we standardized all tissue samples to a final concentration of 100 ng/μL.

Primers for each AeugOBP and AeugCSP (Appendix A) were designed in Primer3Plus [72] and synthesized at T4OLIGO, Irapuato, Mexico. Full-head cDNA was used to standardize amplification conditions for each primer set (Appendix A). All reactions were performed in three technical replicates for each tissue and with the same cDNA concentration (100 ng/μL) at a final volume of 25μL with GoTaq Master Mix (Promega, Madison, WI, USA) on an Arktik Thermal Cycler gradient thermal cycler (Thermo Scientific). PCR results were evaluated by electrophoresis on 1% agarose gels with Red Stain Gel (Biotium, Fremont, CA, USA) and visualized on a Quantum ST5 gel documentation system (Vilber Lourmat, Collégien, France). We sequenced all amplifications of AeugOBP and AeugCSP by Sanger at Macrogen Inc, Seoul, Korea. Sequences were cleaned and assembled in Sequencher v 5.4.6 (Gene Codes, AnnArbor, MI, USA) and compared with the previously characterized in sillico analysis.

### 4.6. Sequence Information

*A. eugenii* chemosensory gene sequences were submitted to the NCBI to obtain accession numbers for AeugOBPs (OP056778-OP056794), and AeugCSP (OP056795-OP056799).

## Figures and Tables

**Figure 1 ijms-24-03406-f001:**
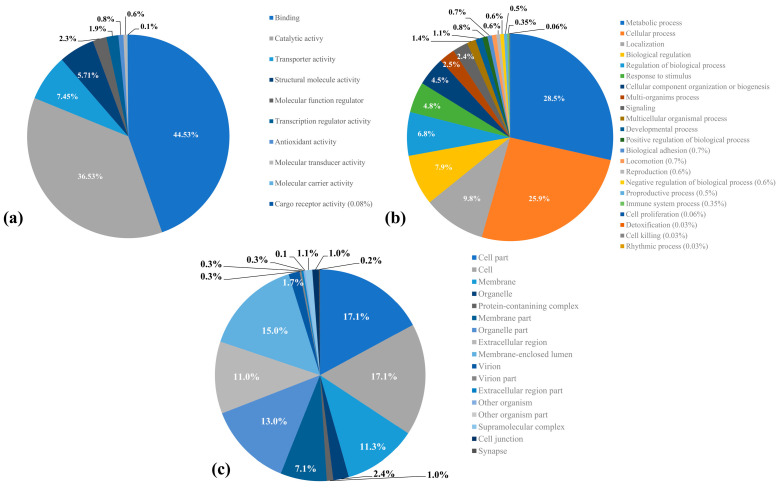
Gene ontology (GO terms) analysis from *Anthonomus eugenii* head transcriptome. The numbers within each graph represent the percentage of genes annotated for each GO term. (**a**) Molecular function level 2; (**b**) biological process level 2; (**c**) cellular component level 2. The percentages were calculated from the total genes annotated per each category.

**Figure 2 ijms-24-03406-f002:**
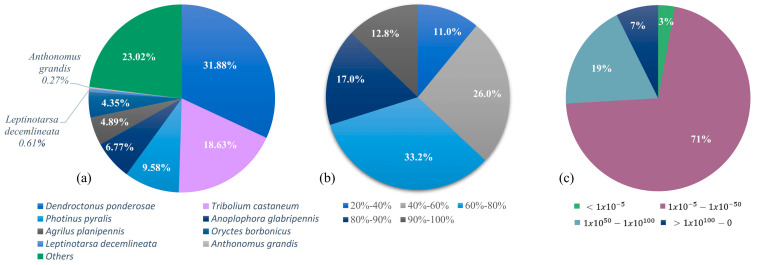
Species annotation obtained from *A. eugenii* head transcriptome mapped against the Insecta-UniProtKB database with BLASTx. (**a**) Species distribution; (**b**) identity percentage distribution; (**c**) e-value distribution.

**Figure 3 ijms-24-03406-f003:**
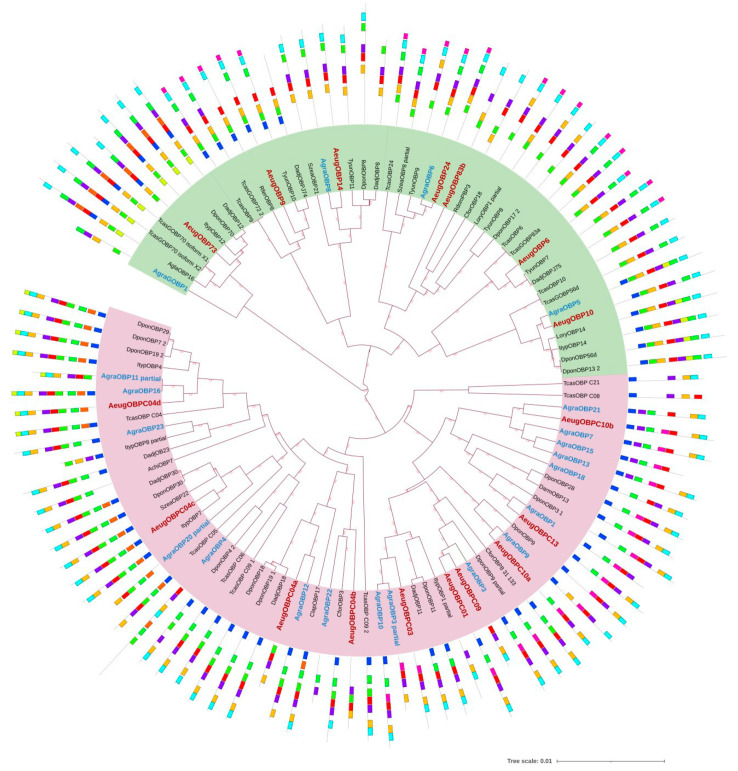
OBPs phylogeny by Bayesian inference. The OBP are divided into two clades: Classics (green) and OBP Minus-C (purple). Motif patterns are related per sequence; the motifs colors correspond to those described in Figure 4. Species: *Anthonomus eugenii* (Aeug), *A. grandis* (Agra), *Rhynchophorus ferrugineus* (Rfer), Rhyzopertha dominica (Rdom), *Sitophilus zeamais* (Szea), *Dendroctonus adjunctus* (Dadj), *D. ponderosae* (Dpon), *Ips typographus* (Ityp) *Tribolium castaneum* (Tcas), *Cylas formicarius* (Cfor), *Phyllotreta striolata* (Pstr), *Tomicus yunannensis* (Tyun), *Lissorhoptrus oryzophilus* (Lory), A*noplophora chinensis* (Achi).

**Figure 4 ijms-24-03406-f004:**
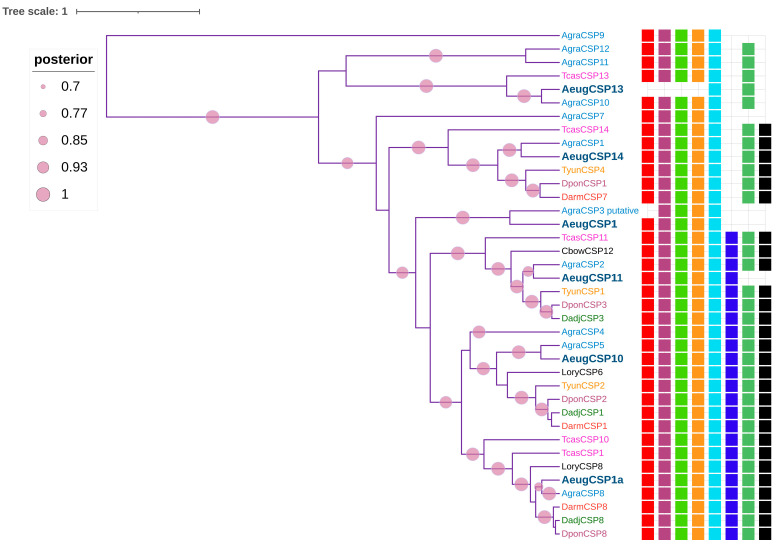
CSPs phylogeny by Bayesian inference. Motif patterns were assigned to each sequence, and motif’s colors correspond to those described in Appendix A. Species: Anthonomus eugenii (Aeug), A. grandis (Agra), Colaphellus bowringii (Cbow), Sitophilus zeamais (Szea), Dendroctonus adjunctus (Dadj), D. ponderosae (Dpon), Tribolium castaneum (Tcas), Tomicus yunannensis (Tyun), Lissorhoptrus oryzophilus (Lory).

**Figure 5 ijms-24-03406-f005:**
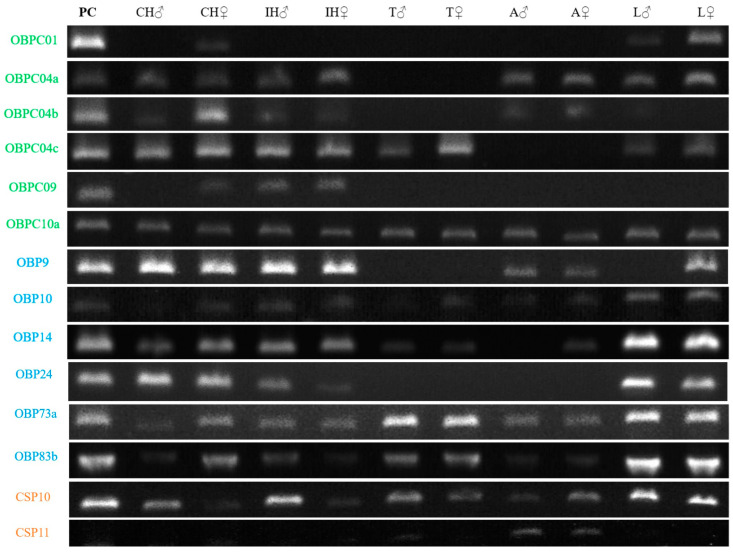
RT-PCR expression pattern by tissue and sex. Different patterns can be observed per tissue and sex. PC: positive control (head pool of both sexes’ complete heads): CH: complete heads (with antennae); IH: incomplete head (without antennae); T: thorax; A: abdomen; L: legs. Green names correspond to Minus-C OBPs, and blue to classic OBPs.

**Table 1 ijms-24-03406-t001:** Functional annotation of AeugOBPs, based on CDD/SPARCLE database.

OBP	PBP-GOBP Superfamily		PhBP Family	
	CDD (cl11600)/PF01395		Smart (smart00708)	
	Range	E-Value	Range	E-Value
AeugOBPC01	21–128	2.84 × 10^−9^	37–129	1.28 × 10^−5^
AeugOBPC03	34–98	5.24 × 10^−4^	30–84	7.50 × 10^−3^
AeugOBPC04a	1–59	1.65 × 10^−5^	−	−
AeugOBPC04b	20–72	4.38 × 10^−4^	−	−
AeugOBPC04c	18–127	1.61 × 10^−14^	32–127	2.64 × 10^−8^
AeugOBPC04d	15–122	1.70 × 10^−22^	29–122	7.99 × 10^−12^
AeugOBP6	1–108	3.18 × 10^−15^	17–108	1.98 × 10^−10^
AeugOBP9	25–137	2.12 × 10^−12^	40–137	8.01 × 10^−7^
AeugOBPC09	1–63	2.49 × 10^−6^	1–63	1.51 × 10^−5^
AeugOBPC10a	10–98	5.11 × 10^−5^	−	−
AeugOBPC10b	79–173	1.19 × 10^−10^	79–173	8 × 10^−10^
AeugOBP10	21–128	5.05 × 10^−24^	34–127	4.62 × 10^−14^
AeugOBP13	25–134	1.84 × 10^−8^	70–133	1.90 × 10^−3^
AeugOBP14	20–134	1.91 × 10^−14^	35–135	8.27 × 10^−7^
AeugOBP24	23–127	1.71 × 10^−7^	59–130	1.52 × 10^−8^
AeugOBP73a	35–144	2.77 × 10^−5^	30–145	5.08 × 10^−3^
AeugOBP83b	16–125	7.50 × 10^−11^	31–125	2.92 × 10^−6^

**Table 2 ijms-24-03406-t002:** Functional annotation of AeugCSPs, based on CDD/SPARCLE database.

CSP	Insect Pheromone-Binding Family, A10/OS-D			
	CDD (cl04042)		Pfam (PF03392)	
	Range	E-Value	Range	E-Value
AeugCSP1	34–89	1.32 × 10^−32^	34–89	1.32 × 10^−32^
AeugCSP1a	-	-	34–127	1.08 × 10^−47^
AeugCSP10	20–110	2.73 × 10^−52^	20–110	2.73 × 10^−52^
AeugCSP11	30–107	5.18 × 10^−44^	30–107	5.18 × 10^−44^
AeugCSP13	10–61	3.19 × 10^−18^	-	-
AeugCSP14	39–130	1.44 × 10^−48^	39–130	1.44 × 10^−48^

## Data Availability

Not applicable.

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
