# Peer review of "Odorant-Binding and Chemosensory Proteins in *Anthonomus eugenii* (Coleoptera: Curculionidae) and Their Tissue Expression"

_ijms, 2023, doi:10.3390/ijms24043406_

Round 1
Reviewer 1 Report
In the present study, Peredes et al tried to explore OBPs and CSPs in the A. eugenii insect. They used transcriptome analysis and bioinformatic tools for this purpose and successfully found relevant genes. Although they provide basic reporting on OBPs and CSPs, that might be helpful in future studies to design environmentally friendly tactics to control A. eugenii. As this study mainly relies on transcriptome data, the description of transcriptome analysis is inadequate and must be mentioned in detail. Therefore, I have low confidence in this manuscript if the required information is not provided clearly. I want to recommend a major revision before acceptance of the manuscript.
The title is long and provides unnecessary information like tissue expression, head, etc. I would suggest changing the title.
Materials and methods
Subsection 4.1
I could not find the number of replicates information in the whole section, and the authors must provide this information. Moreover, do 25 heads of male or female represent one replicate is not clear?
There is no description of the concentration average of Total RNA obtained after the extraction and the quantity of total RNA used for further experiments.
How many libraries were generated and sequenced is not clear in methods, results or even in the supplementary information
There is less or no information about how this study will be helpful to control A. eugenii, and I suggest adding at least 1-2 paragraphs about this.
Reviewer 2 Report
In this paper, Lechuga-Paredes et al. reported the head transcriptome of an economically important pest Anthonomus eugenii. They identified 17 OBPs and 6 CSPs, and analyzed the expression patterns of 12 OBPs and 3 CSPs in different tissues by RT-PCR. The research is generally fine, and manuscript is reasonably well prepared. My comments are given as follows.
1) It is well accepted that the major olfactory organs in insects include antennae, mouthparts, tarsi, and ovipositor; a large number of OBP and CSP genes are mainly or specifically expressed in these organs. I don't know why the authors sequenced the head transcriptomes. This is perhaps the reason for the limited amount of OBPs and CSPs identified in this study.
2) The authors identified 17 OBPs and 6 CSPs. However, they only test the expression profiles of 12 OBPs and 3 CSPs. Please explain the reason.
3) In RT-PCR experiments, the authors used complete head (with antennae) and incomplete head (without antennae) to screen antenna-specific genes. However, I think it is better to collect antennae and use antennal cDNA in RT-PCR.
4) Does Anthonomus eugenii produces sex pheromone? It would be better to discuss the potential role of sex-biased OBPs and CSPs in recognition of sex pheromones.
5) Some typos:
Line 28, AeugOBPs and AeugCSPs
Line 475, A10/OS-D
Line 492, D. ponderosae
Line 504, host plant odor
Round 2
Reviewer 1 Report
I am not satisfied with the response to my comments by the authors. Still, there is no information on the number of replicates used for transcriptome. It is important to show that enough replicates were used to avoid false positive results in the transcriptomic study. Moreover, the authors changed the number of males and females from which tissues were dissected but did not mention whether that number indicates one replicate or more. I would also like to see the transcriptomic data of replicates. Please share it in the supplementary section or in response to my comments.
Round 3
Reviewer 1 Report
A minimum of three replicates is required for a robust study as less than three replicates might produce false positive results in transcriptomic data, and I believe this is the major issue with this study that should be resolved.
Author Response
To prevent false positives and obtain genes with the structural and functional characteristics of OBPs and CSPs, we included as selection criteria: homology analysis with different databases; gene ontology analysis; multiple alignments to identify the conserved Cys patterns; functional domain search with InterproScan, de novo motif analysis, and phylogenetic analyses. In addition, we selected genes identified in sillico, designed specific primers, and amplified the genes in different tissues. Although it is recommended to use three replicates, our objective with the RNA-Seq data was not to perform a differential expression analysis but to reconstruct a de novo transcriptome in order to mine the genes with matches to OBPs and CSPs, characterize them, validate their identification in sillico and also obtain expression profiles in different tissues to know their specificity.
We believe that the work has a solid basis and meticulous analysis, as well as a correct structuring for publication. Due to time limitations, it would not be possible for us to increase the number of replicates of the sequenced libraries, in addition, we would have to redo the analysis from the construction of the de novo transcriptome. We thank you for your suggestions that have helped to improve this work and if required, we will try to make other replicates for the transcriptome part and then look for its publication